# Changes in Inner Retina Thickness and Macular Sensitivity in Patients with Type 2 Diabetes with Moderate Diabetic Retinopathy

**DOI:** 10.3390/biomedicines11112972

**Published:** 2023-11-04

**Authors:** Ana Boned-Murillo, Guisela Fernández-Espinosa, Elvira Orduna-Hospital, Maria Dolores Díaz-Barreda, Ana Sánchez-Cano, María Sopeña-Pinilla, Sofía Bielsa-Alonso, Isabel Pinilla

**Affiliations:** 1Aragon Institute for Health Research (IIS Aragon), 50009 Zaragoza, Spain; anabomu@hotmail.com (A.B.-M.); guisela.fernandez3@gmail.com (G.F.-E.); eordunahospital@unizar.es (E.O.-H.); lodiba@gmail.com (M.D.D.-B.); anaisa@unizar.es (A.S.-C.); 2Department of Ophthalmology, Lozano Blesa University Hospital, 50009 Zaragoza, Spain; 3Department of Ophthalmology, Infanta Sofía University Hospital, 28703 Madrid, Spain; 4Department of Applied Physics, University of Zaragoza, 50009 Zaragoza, Spain; 5Department of Ophthalmology, Virgen de la Luz Hospital, 16002 Cuenca, Spain; 6Department of Ophthalmology, Miguel-Servet University Hospital, 50009 Zaragoza, Spain; maria.sopena1997@gmail.com; 7Department of Surgery, University of Zaragoza, 50009 Zaragoza, Spain; sofiabielsa@gmail.com

**Keywords:** diabetic retinopathy, microperimetry, neurodegeneration, swept-source OCT, type 2 diabetes mellitus

## Abstract

The increase in diabetic retinopathy (DR) prevalence demonstrates the need for the determination of biomarkers for assessing disease development to obtain an early diagnosis and stop its progression. We aimed to analyse total retinal (RT) and inner retinal layer (IRL) thicknesses in type 2 diabetes mellitus (DM2) patients and correlate these results with retinal sensitivity using swept-source OCT (SS-OCT) and microperimetry. For this purpose, a total of 54 DM2 subjects with moderate diabetic retinopathy (DR) with no signs of diabetic macular oedema (DME) and 73 age-matched healthy individuals were assessed using SS-OCT to quantify retinal thickness in the nine macular areas of the ETDRS grid. Retinal sensitivity was measured via microperimetry with a Macular Integrity Assessment Device (MAIA). The mean ages were 64.06 ± 11.98 years for the DM2 group and 60.79 ± 8.62 years for the control group. DM2 patients presented lower visual acuity (*p* < 0.001) and a thicker RT (260.70 ± 19.22 μm in the control group vs. 271.90 ± 37.61 μm in the DM2 group, *p* = 0.01). The retinal nerve fibre layer (RNFL) was significantly lower in the outer nasal area (50.38 ± 8.20 μm vs. 45.17 ± 11.25 μm, *p* = 0.005) in ganglion cells and inner plexiform layers (GCL+) in DM2. A positive correlation between the LDL-C and RNFL and a negative correlation between HDL-C levels and the inner temporal and central RNFL thickness were detected. The central (*p* = 0.021) and inner nasal (*p* = 0.01) areas were negatively correlated between the RNFL and MAIA, while GCL++ was positively correlated with the outer inferior (*p* = 0.015) and outer nasal areas (*p* = 0.024). Retinal sensitivity and macular RNFL thickness decrease in DM2 patients with moderate DR with no DME, and this study enables an accurate approach to this disease with personalised assessment based on the DR course or stage. Thus, GCL+ and GCL++ thinning may support ganglion cell loss before the RNFL is affected.

## 1. Introduction

The prevalence of diabetic retinopathy (DR) is growing worldwide, and it is estimated to increase to 51% by 2045 [1]. It is reported to be the main cause of blindness in the active population of developed countries, especially related to diabetic macular oedema (DME) or severe visual deficits secondary to proliferative DR [2]. Thus, it is critical to investigate and identify biomarkers to assess disease development. This would therefore improve its management and stop its progression, reducing its morbidity and mortality.

DR is a microvasculopathy that induces changes in the inner retina, affects ischaemia, and increases blood-retina barrier permeability. Both vascular changes and diabetic neurodegeneration (DN) occur early in the disease, first affecting the retinal ganglion cell bodies and their dendrites, which can be detected as diffuse thinning of the ganglion cell layer (GCL) and inner plexiform layer (IPL) [3]. Axons may undergo apoptosis, triggering thinning of the retinal nerve fibre layer (RNFL). The ganglion cell complex (GCC), formed by the GCL, IPL, and RNFL, may also show a reduction in thickness [3,4]. These variations have been described as inner retinal layer (IRL) thickness changes in the absence of any DR modifications prior to the appearance of diabetic vascular signs detectable on fundus examination by an ophthalmologist [2].

The structural changes in DR can be quantitatively analysed with new imaging techniques, mainly high-speed non-invasive optical coherence tomography (OCT), allowing the evaluation of the thickness and volume of the different retinal layers and the choroid [5,6,7]. This method is more objective and accurate than ophthalmoscopy for identifying early structural alterations in the diabetic retina. DN and RNFL loss in turn cause functional modifications in several diagnostic tests that can be detected prior to the appearance of the first manifestations of DR, including electroretinogram [8], contrast sensitivity, dark adaptation, and microperimetry alterations [9,10,11]. The latter is especially useful for evaluating macular functionality due to its high sensitivity and specificity.

The purpose of this study was to measure macular changes in the IRL thicknesses studied using swept-source OCT (SS-OCT) in type 2 diabetes mellitus (DM2) patients with moderate DR and without DME compared to those of healthy control individuals and to correlate them with retinal function evaluated via microperimetry to identify markers of retinal neurodegeneration prior to the appearance of DR.

## 2. Materials and Methods

### 2.1. Study Design

A total of 127 eyes were included in our study. Of these, 54 eyes were from DM2 patients, and 73 eyes were from healthy subjects. All eyes were evaluated at the Ophthalmology Department in the Lozano Blesa University Hospital in Zaragoza, Spain, from October 2021 to June 2022. All subjects underwent a complete ophthalmological exam in a single visit. The DM2 patients, which constituted Group 1, demonstrated a level 43 on the early treatment diabetic retinopathy study (ETDRS) classification [12], which corresponds to moderate DR, and were without DME. Group 2 consisted of healthy subjects with no previous history of ocular or systemic diseases. The study was approved by the local Ethics Committee (Clinical Research Ethics Committee of AragonCEICA PI19/252) and adhered to the tenets of the Helsinki Declaration. Each subject signed the informed consent form.

Exclusion criteria for both groups were amblyopia or best corrected visual acuity (BCVA) lower than 20/40 in the Snellen scale, spherical equivalent (SE) above ±5.50 diopters (D) or 3.00 D of astigmatism, any changes in pupillary reflex or ocular motility, intraocular pressure (IOP) over 20 mmHg or findings that suggested glaucoma, other macular diseases with macular impairment that lead to scotomas that affect vision, history of ocular surgery, corneal degeneration or lens opacity, the spectrum of vision loss related to dry eye disease, or the impossibility of collecting a good quality OCT profile. The electronic medical histories of the participants were deeply reviewed, looking for personal systemic pathologies. Patients with uncontrolled arterial hypertension, diagnosis of any metabolic syndrome, different form of DM, other systemic pathologies, or diagnosis or signs of neurologic or neurodegenerative disorders (multiple sclerosis, Alzheimer’s and Parkinson’s disease, bipolar disorder, etc.) were also excluded as patients with hormone replacement therapies.

### 2.2. Study Protocol

All subjects underwent a complete ophthalmological exam in a single visit. For statistical purposes, BCVA was recorded with the 100% contrast ETDRS test as measured with the logarithm of the minimum angle of resolution (LogMAR). Axial length (AL) was calculated using an Aladdin KR-1 W Series optical biometry system (Topcon Corporation, Tokyo, Japan) as the average of 5 measurements and expressed in mm. IOP was measured via Goldmann tonometry. The eye fundus was examined using Clarus widefield retinography (Clarus 700^®^, Carl Zeiss Meditec AG, Jena, Germany) images.

Endocrinological data were collected from DM2 patients; this data included the number of years since diagnosis, glycaemic control measured with glycosylated haemoglobin (HbA1c), lipid profile, renal function parameters, and medication.

OCT and widefield retinography were used to exclude subclinical and clinical retinal neovascularization and DME. All OCT images were acquired by the same investigator (ABM) using DRI-Triton SS-OCT (deep range imaging) (Topcon Corporation, Tokyo, Japan). The software version was IMAGEnet 6 Version 1.22.1.14101^®^. Images whose quality scale (0 to 100) was lower than 60 were excluded. The different retinal layer and retinal thickness (RT) values were expressed in micrometres (μm) in the different sectors of the ETDRS grid with the 3D Macula protocol (Figure 1). The central (C) area is a circle of 1 mm in diameter, surrounded by the parafoveal or inner ring, which is 3 mm in diameter. This area was divided into 4 quadrants: inner superior (IS), inner temporal (IT), inner nasal (IN), and inner inferior (II). The outer or perifoveal circle had a diameter of 6 mm and was divided into 4 quadrants: outer superior (OS), outer temporal (OT), outer nasal (ON), and outer inferior (OI).

To evaluate retinal sensitivity, third generation microperimetry (the Macular Integrity Assessment Device (MAIA); Topcon Corporation, Tokyo, Japan) was used for a complete evaluation with a 4–2 complete threshold strategy. To compare the MAIA and OCT thickness results, the sensitivity points generated with the microperimetry were divided into sectors. For an emmetropic eye, the MAIA 1° is equivalent to a circle with a diameter of 0.6 mm, the MAIA 3° is equivalent to a circle with a diameter of 1.8 mm, and the MAIA 5° is equivalent to a circle with a diameter of 3 mm. In the central ETDRS ring, we included the centre point and the 1° sensitivity points (0.6 mm diameter). In the 3 mm ETDRS ring, the 3° and 5° sensitivity points were located (diameters of 1.8 and 3 mm, respectively) in the 3 mm circle of the ETDRS grid [9]. Therefore, the average of the retinal sensitivity thresholds calculated for the 1° MAIA corresponds to the central ETDRS circle, and the thresholds of the 3° and 5° circles correspond to the ETDRS inner or parafoveal ring (mean of 6 sensitivity points/quadrant; Figure 1). The fixation was inspected manually.

The data were collected and exported to an Excel database (Microsoft Corporation, Redmond, WA, USA).

### 2.3. Statistical Analysis

The values of the variables of interest for each patient were collected in an Excel database ver. 2016 (Microsoft Corp., Redmond, WA, USA). For the statistical analysis, the Statistical Package for the Social Sciences software (SPSS version 20.0, SPSS Inc., IBM Corporation, Armonk, NY, USA) was used. First, a descriptive cross-sectional analysis of the sample with demographic variables and clinical characteristics was performed. Data normality was analysed with the Kolmogorov-Smirnov test. The parameters did not have a normal distribution, so differences between groups were analysed with the Mann–Whitney U test for independent samples, and the Spearman’s rank correlation coefficient test was conducted for bivariate analysis to correlate the variables of interest. A value of *p* < 0.05 indicated statistical significance for all the analyses.

The sample size was calculated based on a preliminary study carried out by our group using the Epidat software (version 4.2.0.0). All calculations were performed using a two-sided test with an α risk of 5% (95% confidence level) and a β risk of 20% (80% power). A standard deviation of retinal thickness of 30 μm was estimated, wanting to detect differences of at least 16 μm, and setting a ratio between independent sample sizes of 1.5, a minimum sample size of 71 and 47 subjects per group was determined.

## 3. Results

### 3.1. Demographics

A descriptive cross-sectional analysis of the sample determined that the mean age of the DM2 group was 64.06 ± 11.98 years (42–86 years) and that of the control group was 60.79 ± 8.62 years (42–83 years), with no differences between groups (*p* = 0.082). Regarding sex distribution, 20.4% and 39.7% were females and 79.6% and 60.3% were males in the DM2 and control groups, respectively. The mean time since DM2 diagnosis was 2.50 ± 2.88 years (0–11 years). The patients had adequate metabolic control of their disease, with a mean HbA1c of 7.58 ± 1.29%. Table 1 presents glycaemic, lipid, and renal function values.

BCVA reached significantly lower levels in the DM2 group (*p* = 0.001). No differences were found between groups in AL (*p* = 0.075), SE (*p* = 0.110), or IOP (*p* = 0.676). The values are presented in Table 1.

### 3.2. OCT: Total Retina and IRL Thickness Assessment

Analysis of retinal thickness with SS-OCT revealed statistically thicker RT in the OT area in the DM2 group than in the healthy group (260.70 ± 19.22 μm in the healthy group vs. 271.90 ± 37.61 μm in the DM2 group, with *p* = 0.010). Additionally, the analysis of different protocols revealed a significant decrease in the RNFL,specifically in the ON area. Thickness thinning was also observed in the IS, IT, and IN quadrant of the GCL+ protocol (GCL+IPL) ; and in the temporal quadrants as in ON and IS of the GCL++ protocol (ILM-IPL/INL (GCC), which corresponds to the GCL+ plus the RNFL thickness) in the DM2 group, as shown in Figure 2.

### 3.3. MAIA Retinal Sensitivity Assessment

According to MAIA microperimetry, the retinal sensitivities were significantly higher in the control group than in the DM2 group, and there were differences in macular integrity (77.82 ± 28.04 vs. 64.84 ± 30.02; *p* = 0.005) and total mean threshold (24.45 ± 3.63 vs. 26.69 ± 2.26; *p* < 0.001), as shown in Table 2.

### 3.4. Structural and Functional Correlations

Correlations between OCT findings (RT and RNFL thickness) and other parameters, including age, years of DM evolution, and glycaemic control, were studied using Spearman’s rank correlation coefficient test. Significant negative correlations were found between age and the outer ETDRS ring areas of RT, GCL +, and GCL++ (*p* < 0.05). Additionally, the central ring for total RT and age were significantly positively correlated (cc = 0.327; *p* = 0.018), as shown in Figure 3. No significant correlations were observed between RNFL thickness and MAIA sensitivity and age, disease evolution, or HbA1c levels, as indicated in Figure 3.

Structural and functional correlations were demonstrated in the DM2 group. MAIA sensitivities and RNFL thickness were negatively correlated (r = −0.324, *p* = 0.021, and r = −0.443, *p* = 0.001, in the C and IN areas, respectively). In addition, positive and significant correlations between MAIA sensitivities and GCL++ thickness were detected (r = 0.339, *p* = 0.015, and r = 0.316, *p* = 0.024, in the OI and ON areas, respectively). No significant correlation was found between MAIA sensitivities and GCL+ thickness in any of the analysed sectors, as shown in Figure 4.

### 3.5. RNFL and GCL Thickness Correlations

The correlations between RNFL and GCL thicknesses and the metabolic characteristics of DM2 patients were evaluated. There were different significant correlations, as shown in Appendix A.

## 4. Discussion

Our results suggested structural and functional changes in patients with moderate DR without DME. Even though early changes in preclinical DR remain elusive, numerous studies have demonstrated early neurodegeneration prior to DR manifestations, with an impairment of the neurovascular unit—including neurons, glia, and vasculature [13]—and structural and functional changes—including RNFL thinning, prior to the appearance of other DR signs, which may be considered a marker of this disease [14]. Neurodegeneration will progress as soon as vascular lesions appear with a higher impairment of retinal neurons. Our goal is to achieve early detection of this neuronal damage with simple, non-invasive techniques and descriptive markers.

The inner retina is highly susceptible to metabolic stress because of its high metabolic demand and relatively lower blood perfusion. Chronic hyperglycaemia is thought to affect retinal ganglion cells, altering their function and leading to their impairment and death, with consequent GC-IPL and RNFL thickness loss [15]. Retinal neural cell apoptosis could be related to neurofilament accumulation in the RNFL, secondary to changes in retrograde axonal transport, a rise in extracellular glutamate levels due to the impairment of Müller cells, toxicity to neurons, an increase in neurotoxic factors [16], and reactive changes in microglia [8]. Moreover, the production of erythropoietin and inflammatory mediators associated with increased levels of vascular endothelial growth factor may cause vascular damage and impair the ability to regulate local blood flow [16,17].

We found total RT thinning at the parafoveal ring, with a reduction in both GCL and IPL thickness, which has been demonstrated by previous studies [15,16]. This suggests that ganglion cells are one of the most susceptible to neurodegenerative and vascular effects in DM patients. In the same manner that has already been demonstrated in several neurodegenerative diseases, such as glaucoma, multiple sclerosis, Parkinson’s disease, and Alzheimer’s disease, neurodegeneration was demonstrated via macular RNFL thinning [18]. Specifically, we found significant differences in the ON quadrant compared to healthy control subjects. These findings were similar to those of other researchers, such as Carpineto et al. [19] and Jia et al. [20]. There are few previous studies evaluating macular RNFL thinning that focus on the peripapillary RNFL (pRNFL) thickness, and the results and the affected quadrants of these studies varied [19,20,21,22,23]. These discrepancies may be explained by differences in age, glycaemic status, DR severity, DM duration, and comorbidities among the studied populations, in addition to using different imaging modalities and programmes to measure pRNFL.

The study of GCL and RT thickness revealed a decrease with age in the perifoveal ETDRS areas. Although GCL+ and GCL++ presented a significant negative correlation in the outer ring (OS, OT, OI, and ON), there was no significant correlation between the RNFL and age, which may support ganglion cell loss before the RNFL is affected. Previous researchers, such as Rasheed et al., described an association between DR and diabetic neuropathy. In this study, patients with neuropathy showed significant thinning in the GC-IPL earlier than in the pRNFL. Similarly, Srinivasan et al. suggested that ganglion cell loss is a predictor of diabetic peripheral neuropathy [22].

We could not confirm RNFL thickness deterioration with DR progression, diabetes progression time, or glycaemic variability. Contrary to our results, other researchers, such as Shi et al. [24] and Dashmana et al. [23], demonstrated that thinning of the RNFL in the superior quadrant was significantly correlated with diabetes duration, suggesting that the thinning of this sector could be the primary structural change in DR. Carpineto et al. [19] described that both the average (r = −0.236, *p* = 0.033) and inferior quadrants (r = −0.216, *p* = 0.049) of RNFL thickness were negatively correlated with HbA1c levels.

We also analysed systemic risk factors (including serum lipid levels, blood pressure, and glucose levels) to determine their involvement in diabetic retinal changes. We excluded patients with poor control of their arterial blood pressure or a diagnosis of metabolic syndrome to avoid confounding factors. Our results revealed a significant positive correlation between LDL-C and RNFL (r = 0.285, *p* = 0.037). Cholesterol is an important component of myelin and regulates membrane fluidity and signalling proteins. An increase in total cholesterol (TC) and LDL has been related to adverse effects on the RNFL [25]. Additionally, significant negative correlations between HDL-C levels and RNFL thickness (r = −0.362, *p* = 0.007, and r = −0.292, *p* = 0.032, in the IT and C global sectors, respectively) and C global GCL+ and GCL++ (r = −0.379, *p* = 0.007, and r = −0.383, *p* = 0.005, respectively) were found by other researchers, such as Shi et al. (r = −0.223, *p* = 0.042) [24]. HDL cholesterol has been associated with a thinning of the RNFL in multiple sclerosis (r = −0.15, *p* = 0.008). This has been postulated to be related to blood–brain barrier breakdown and the extravasation of immune cells through the vascular endothelium in other neurodegenerative diseases [26]. The involvement of the HDL pathway in visual dysfunction has been determined in age-related macular degeneration and supports the plausibility of the associations we have identified [27].

We also found that triglyceride (TG) levels were positively correlated with GCL+ thickness in the OS, II, and IS areas (r = 0.296, *p* = 0.037; r = 0.307, *p* = 0.030; and r = 0.296, *p* = 0.037, respectively), which to our knowledge has not been determined previously in the literature. Similarly, previous studies have described a positive correlation between TG levels and INL thickness in DM1 subjects (r = 0.48, *p* = 0.011) [28]. This may differ from the results of Shi et al., which revealed a negative correlation between RNFL thickness in the inferior quadrant and TG levels (r = −0.232, *p* = 0.035) [24]. These findings may promote future lines of work that seek to determine the role of serum lipids in retinal neurodegeneration prevention.

Contrary to our expectations, creatinine levels were positively correlated with the central area of the RNFL and the GCL+ and GCL++ thicknesses (r = 0.452, *p* = 0.001; r = 0.475, *p* < 0.001; and r = 0.448, r = 0.001, respectively) and negatively associated with filtrate rate (FR) deterioration (r = −0.465, *p* < 0.001; *p* = −0.371, *p* = 0.08; and r = 0.383, *p* = 0.008, respectively). Previous studies have described retinal thinning as a biomarker of renal impairment in patients with diabetes (*p* = 0.009) [29]. Srivastav et al. found positive correlations between RNFL thinning and increases in serum urea and creatinine levels in patients with different DR stages [30].

Regarding the relationship between structural and functional findings, we correlated macular ETDRS grid areas with the corresponding microperimetry points. We did not contemplate GC displacement from their receptive, as evidence is not strong enough due to great variability between subjects [31,32]. Although decreased retinal sensitivity could be reflected by the significantly worse BCVA, the correlations support our results. We observed negative correlations between the RNFL in the IN and in the C areas and retinal sensitivity measured with the MAIA, which were not present when evaluating the GCL+. These findings are similar to those of Orduna et al. [11]. Clinical diabetes may critically affect patients’ vision, even producing permanent visual acuity loss. The diagnosis and follow-up of these patients require adequate functional tests, and microperimetry obtains an exact fundus-related quantification of retinal sensitivity [10]. There is substantial evidence of reduced microperimetric sensitivity in patients with diabetes either with or without DR compared to healthy subjects without diabetes, as evaluated using different microperimetry equipment, including Optos OCT/SLO/microperimeter [33], MP1/Nidek Technologies [23], MP-3/Nidek [34], or, as in our study, the Macular Integrity Assessment Device (MAIA). Contrary to our results, no significant correlation between retinal sensitivity and retinal thickness was found by previous researchers, such as Chai et al. [14] or Rohrschneider et al. [9].

However, we have to consider the limitations of microperimetry, which include sources of variability such as fatigue and learning. Differences between right and left eye parameters related to fatigue have been described. This is because the duration of the test exceeds 5 min, and in convention, the right eye is analysed first [35]. This could be reduced by testing only the study eye or if we perform it before the rest of the tests. Regarding microperimetry learning tests, it is not clear if performing a theme is recommended, as the duration of learning effects is unknown [35].

## 5. Conclusions

In conclusion, our results suggest that the structure (total retinal, GC-IPL, and RNFL thicknesses) and function (retinal sensitivity) of the retina in patients with moderate diabetes display some changes. We demonstrated a correlation between the RNFL and retinal sensitivity measured with the MAIA, and the findings were not present when evaluating the GCL+. The diagnosis of GCL and RNFL thinning in DM2 patients without DR prior to the appearance of other DR signs would enable an accurate approach to this growing disease, with personalised assessment based on the DR course or stage. Future studies require a larger sample size, which should include patients in different stages of the disease and both DM1 and DM2 patients. Systemic risk factors are expected to be involved in retinal neurodegeneration in DM2 patients. However, the role of these factors remains largely unknown, and future lines of work should be promoted to determine it. 

## Figures and Tables

**Figure 1 biomedicines-11-02972-f001:**
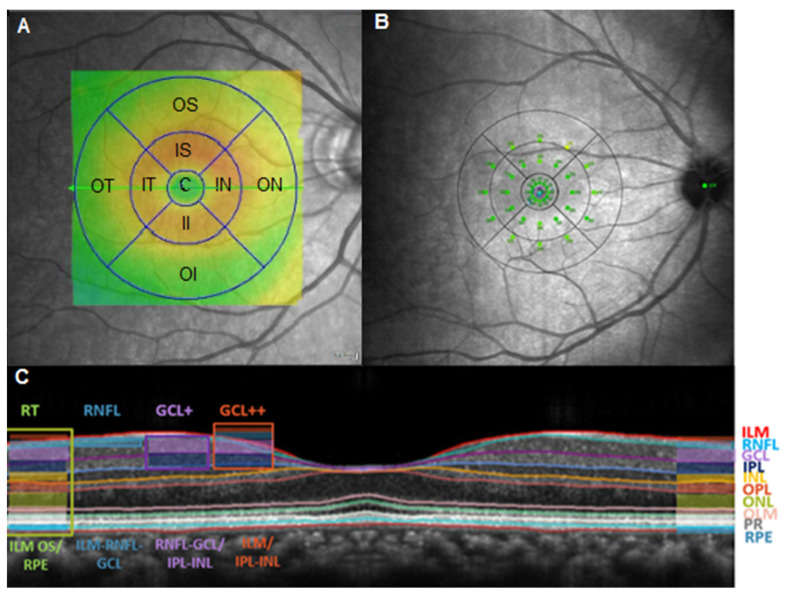
(**A**) Grid of macular sectors for the DRI-Triton Swept-Source OCT (SS-OCT) for the 9 areas of the ETDRS grid in a right eye (OS, outer superior; OT, outer temporal; OI, outer inferior; ON, outer nasal; IS, inner superior; IT, inner temporal; II, inner inferior; IN, inner nasal; and C, central). (**B**) Mean retinal sensitivity in dB measured with the MAIA microperimeter and correlated with 9 areas of the ETDRS grid. (**C**) DRI-Triton (SS)-OCT profile showing the protocols used in the study: total retina (from the internal limiting membrane (ILM) to the boundary between the retinal pigment epithelium (RPE) and the photoreceptor layer (OS/RPE limit)), GCL+ protocol (from the internal boundary of the ganglion cell layer (GCL; line RNFL/GCL) up to the external limit of the IPL (the IPL/INL line)), and GCL++ protocol (from ILM to the IPL/INL line (GCC)).

**Figure 2 biomedicines-11-02972-f002:**
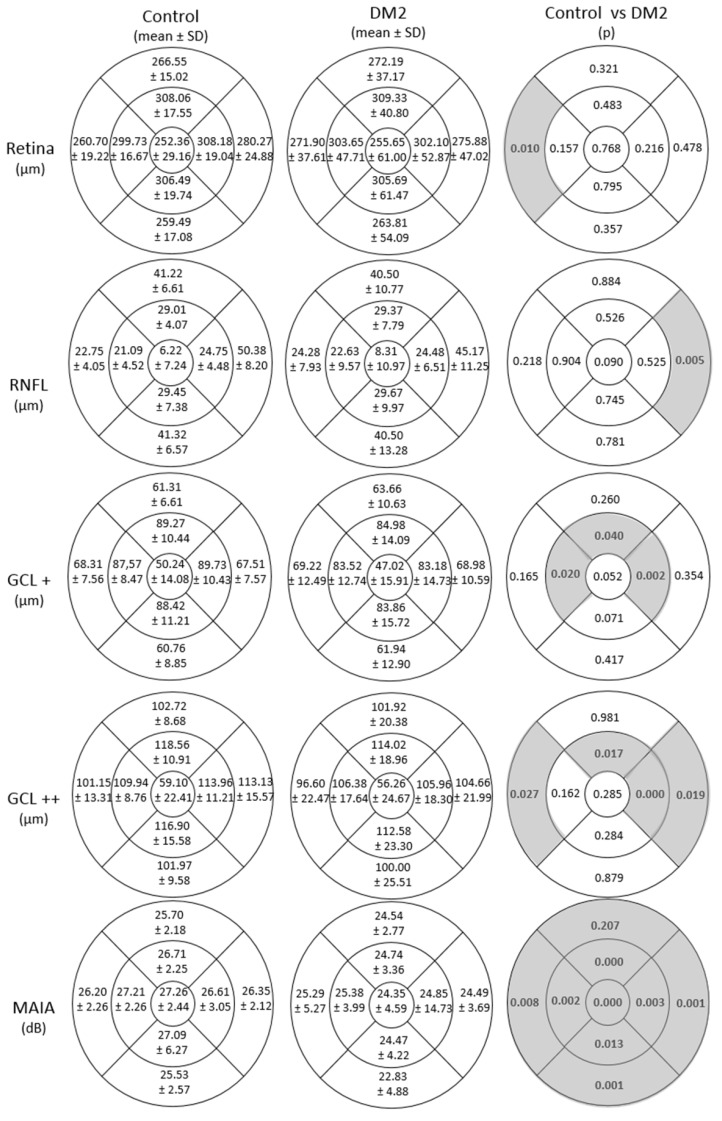
Mean ± standard deviation (SD) of total retina, retinal nerve fibre layer (RNFL), GCL+ protocol (GC-IPL) and GCL++ protocol (ILM-IPL/INL (GCC)) thicknesses measured using DRI-Triton SS-OCT and mean retinal sensitivity in dB measured with the MAIA microperimeter in patients with type 2 diabetes mellitus (DM2) and in healthy control subjects and their comparison (*p* value) in the 9 areas of the early treatment diabetic retinopathy study (ETDRS) grid (OS, outer superior; OT, outer temporal; OI, outer inferior; ON, outer nasal; IS, inner superior; IT, inner temporal; II, inner inferior; IN, inner nasal; and C, central), where temporal quadrants are represented on the left and nasal quadrants are represented on the right. Statistically significant differences (*p* < 0.05) are marked in bold with a grey background.

**Figure 3 biomedicines-11-02972-f003:**
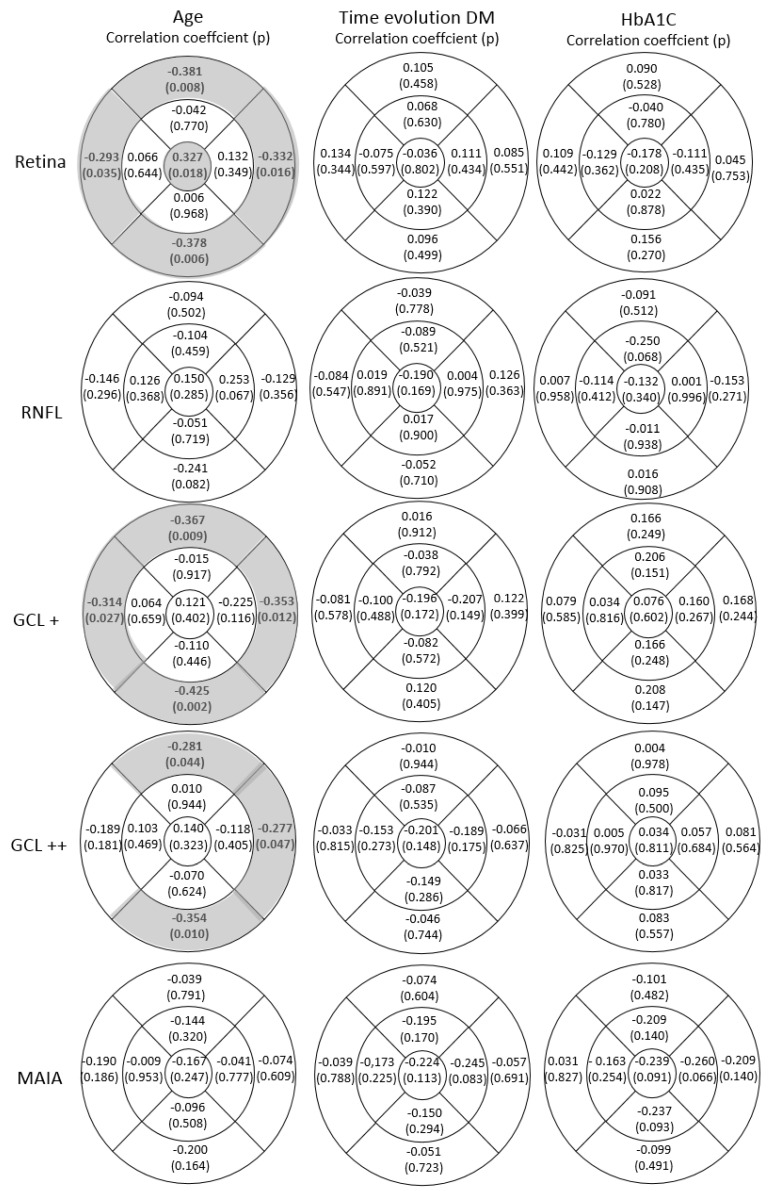
Correlation coefficients and statistical significance (*p* value) of retinal, retinal nerve fibre layer (RNFL), GCL+ (GC-IPL), GCL++ protocol (ILM-IPL/INL [GCC]) thickness, and mean retinal sensitivity in dB measured with the MAIA microperimeter represented in the nine areas of the early treatment diabetic retinopathy study (ETDRS) grid (OS, outer superior; OT, outer temporal; OI, outer inferior; ON, outer nasal; IS, inner superior; IT, inner temporal; II, inner inferior; IN, inner nasal; and C, central) and where temporal quadrants are represented on the left and nasal quadrants are represented on the right, with age, time of DM evolution, and glycosylated haemoglobin (HbA1c) levels (%) in DM2 patients. The values that reached statistical significance (*p* < 0.05) are shown in bold with a grey background.

**Figure 4 biomedicines-11-02972-f004:**
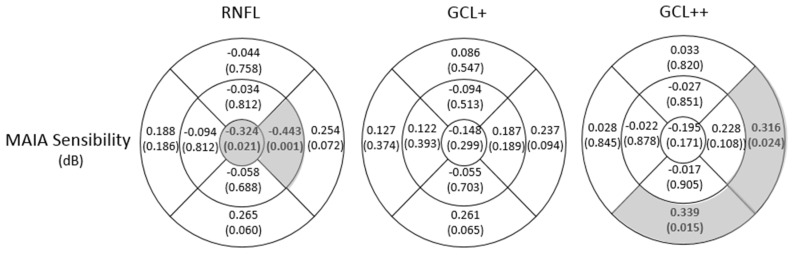
Correlation coefficients and statistical significance (*p* value) between MAIA retinal sensitivity and retinal nerve fibre layer (RNFL), GCL+ (GC-IPL), and GCL++ protocol (ILM-IPL/INL [GCC]) in DM2 patients in the nine areas of the early treatment diabetic retinopathy study (ETDRS) grid (OS, outer superior; OT, outer temporal; OI, outer inferior; ON, outer nasal; IS, inner superior; IT, inner temporal; II, inner inferior; IN, inner nasal; and C, central), where temporal quadrants are represented left and nasal quadrants are represented right. The values that reached statistical significance (*p* < 0.05) are shown in bold with a grey background.

**Table 1 biomedicines-11-02972-t001:** Mean, standard deviation (SD), and statistical significance (*p* value) of demographics; best corrected visual acuity (BCVA) in the LogMAR scale; spherical equivalent (SE) in diopters (D); axial length (AL) in mm; and intraocular pressure (IOP) in mmHg between the control and type 2 diabetes mellitus (DM2) groups as well as the metabolic characteristics of DM2 patients related to the duration and metabolic control of the disease. Abbreviations: HbA1c, glycosylated haemoglobin; HDL, high-density lipoprotein; LDL, low-density lipoprotein; TG, triglyceride; GF, glomerular filtration; and SD, standard deviation. HbA1c values are expressed in a percentage; cholesterol, TG, and creatine values are expressed in mg/dL; and GF is expressed in mL/min. Differences that reached statistical significance (*p* < 0.05) are shown in bold.

	Control Group		DM2 Group		
	Mean	SD	Mean	SD	*p*
Age (years)	60.79	8.62	64.06	11.98	0.082
Sex (female-male %)	39.7–60.3		20.4–79.6		
Time from diagnosis (years)			2.50	2.88	
HbA1c (%)			7.58	1.29	
BCVA (LogMAR)	0.04	0.05	0.12	0.17	**<0.001**
SE (D)	0.03	1.58	0.37	1.70	0.110
AL (mm)	23.73	1.46	23.23	0.84	0.080
IOP (mmHg)	15.30	2.89	14.76	2.49	0.676
Disease progression time (years)			2.50	2.88	
HbA1c (%)			7.58	1.29	
Cholesterol (mg/dL)			148.04	33.18	
HDL (mg/dL)			47.83	15.21	
LDL (mg/dL)			71.47	23.09	
TG (mg/dL)			122.24	51.71	
GF (mL/min)			73.57	20.52	
Creatine (mg/dL)			1.05	0.49	

**Table 2 biomedicines-11-02972-t002:** Mean ± standard deviation (SD) of the sensitivity of the variables was measured with MAIA microperimetry in diabetes mellitus (DM2) patients and in healthy control subjects and was compared (BCEA, bivariate contour ellipse area). Statistically significant values (*p* < 0.05) are shown in bold.

Retinal Sensitivity (dB)			
	Control	DM	Control vs. DM
	Media ± SD	Media ± SD	*p*
Macular integrity	64.84 ± 30.02	77.82 ± 28.04	**0.005**
Average threshold	26.69 ± 2.26	24.45 ± 3.63	**<0.0001**
Fixation stability P1	88.25 ± 13.03	77.96 ± 26.02	0.162
Fixation stability P2	96.78 ± 4.71	89.26 ± 17.31	**0.016**
BCEA 63 area	1.82 ± 1.93	4.33 ± 6.65	0.121
BCEA 63 angle	2.39 ± 62.12	9.20 ± 50.55	0.637
BCEA 95 area	45.35 ± 5.79	12.84 ± 20.03	0.142
BCEA 95 angle	2.39 ± 6.11	4.64 ± 50.25	0.956
Fixation loses (%)	4.20 ± 10.71	7.34 ± 17.99	0.432

## Data Availability

A.B.-M. is the guarantor of this work and, as such, had full access to all the data in the study and takes responsibility for the integrity of the data and the accuracy of the data analysis. The datasets used and/or analysed during the current study are available from the corresponding author on reasonable request.

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
