# Peer review of "Changes in Inner Retina Thickness and Macular Sensitivity in Patients with Type 2 Diabetes with Moderate Diabetic Retinopathy"

_biomedicines, 2023, doi:10.3390/biomedicines11112972_

Round 1
Reviewer 1 Report
Comments and Suggestions for Authors
Dear Ana Boned-Murillo and colleagues,
Please find enclosed the comments on your manuscript entitled “Changes in inner retina thickness and macular sensitivity in diabetic 2 patients with moderate diabetic retinopathy”
The manuscript is unfortunately in need of a thorough revision. While the intension of the work is clear, the manuscript reads like a draft.
Beside intensive text editing, it is very important to open the text to a wider audience. The overuse of abbreviations and technical terms limits currently the readership to a rather narrow circle.
Abstract: the abstract needs to start by giving the relevant context defining the research need (i.e. why it is important to know about retina thickness and macular sensitivity in diabetic patients? The abstract reads currently like bullet points and the key information: why was the research done, what is the hypothesis/research need, how was it done, what are the key findings and what is their impact needs to be presented in a narrative way.
Figure 2 needs to be enlarged as it very difficult to read. What are the units?
Results: each result section needs to start with a brief introduction explaining why the experiments was performed and how it was done. All abbreviations like BCVA, AL, SE, IOP, RNFL, GCL+, GCL++ need to be explained also in the result section. Researchers less familiar with your work would not know what for example BCVA stands for although the differences reached significance.
Table 2 has no units
Discussion: Please remind the reader at the start of this section what the significant findings of the work are. Please make reference to the corresponding display items.
Further experimental work is not required.
Comments on the Quality of English LanguageThorough revision needed.
Author Response
Dear Reviewer,
We appreciate your comments about the interest of this manuscript.
We revised the manuscript according to your suggestion and the last change included in the manuscript is detailed below.
We will try to answer your comments point-by-point.
- Extensive editing of English language required
Manuscript has undergone an extensive English revision as suggested (Please see the attachment).
- Abstract: the abstract needs to start by giving the relevant context defining the research need. The abstract reads currently like bullet points and the key information: why was the research done, what is the hypothesis/research need, how was it done, what are the key findings and what is their impact needs to be presented in a narrative way.
Abstract was rewritten into a narrative way as suggested, to this purpose some sentences were changed: ‘’ ''The increase in diabetic retinopathy (DR) prevalence demonstrates the need for the determination of biomarkers for assessing disease development to obtain an early diagnosis and stop its progression. We aimed to analyse total retinal (RT)…’’ and ‘’Retinal sensitivity and macular RNFL thickness decrease in DM2 with moderate DR with no DME and its study would enable an accurate approach to this disease with personalized assessment based on the DR course or stage’’ and ‘’… Retinal sensitivity and macular RNFL thickness decrease in DM2 with moderate DR with no DME and its study would enable an accurate approach to this disease with personalized assessment based on the DR course or stage. In the same way, GCL+ and GCL++ thinning may support ganglion cell loss before the RNFL is affected.‘’
- Figure 2 needs to be enlarged as it very difficult to read. What are the units?
Figure 2 has been enlarged to be clearly read and the units are detailed in the left side of the image, under each row.
- Results: each result section needs to start with a brief introduction explaining why the experiments was performed and how it was done. All abbreviations like BCVA, AL, SE, IOP, RNFL, GCL+, GCL++ need to be explained also in the result section.
A brief introduction explaining why the experiments were performed and how it was done has been added in each result section as suggested: Demographics ‘’After a descriptive cross-sectional analysis of the sample we determined the mean age of the DM2 group was…’’, OCT: total retina and IRL thickness assessment ‘’Analysis of retinal thickness with SS-OCT revealed statistically significant differences in the OT area…’’.
We have described all the used abbreviations below each table and figure in the result section.
- Table 2 has no units
Table 2 units are indicated in the superior area of the table: ‘’Retinal sensitivity (dB)’’.
- Discussion: Please remind the reader at the start of this section what the significant findings of the work are. Please make reference to the corresponding display items.
Discussing has been modified to remark the finding of the work at the beginning: ‘’ Our results suggested structural and functional changes in patients with moderate DR without DME; these changes included RNFL thinning prior to the appearance of other DR signs, which may be considered a marker for an accurate approach to this disease… Our goal is to achieve early detection of this neuronal damage with simple noninvasive techniques and descriptive markers …’’

Reviewer 2 Report
Comments and Suggestions for Authors The authors of the manuscript ID: biomedicines- 2629805, title: " Changes in inner retina thickness and macular sensitivity in diabetic 2 patients with moderate diabetic retinopathy presented interesting experimental studies with high clinical potential.The authors analyzed total retinal (RT) and inner retinal layers(IRL) thicknesses in type 2 diabetes mellitus (DM2) patients and retinal sensitivity, using swept source OCT(SS-OCT), and microperimetry.
A total tested of 127 eyes, from 54 DM2 subjects with moderate diabetic retinopathy(DR) with no signs of diabetic macular edema(DME) and 73 age-matched healthy individuals were assessed by SS-OCT to quantify retinal thickness in the nine macular areas of the ETDRS grid. The authors showed that retinal sensitivity and macular RNFL thickness decreased in DM2 with moderate DR with no DME. Moreover, GCL+ and GCL++ thinning may support ganglion cell loss before the RNFL is affected.
The manuscript is well prepared, but there are a few errors and requires refinement The introduction is weak, written too concisely, only 4 publications.
In Figure 2 is not visible ground marked in bold
The literature should be prepared in accordance with the requirements of the Biomedicines journal
Author Response
Dear Reviewer,
We appreciate your comments about the interest of this manuscript.
We revised the manuscript according to your suggestion and the last change included in the manuscript is detailed below.
We will try to answer your comments point-by-point.
- The introduction is weak, written too concisely, only 4 publications.
As suggested, introduction has been rewritten, changing some sentences (‘It is reported to be the main cause of blindness in the active population of developed countries, especially related to diabetic macular oedema (DME) or severe visual deficits secondary to proliferative DR [2]. Thus, it is critical to investigate and identify biomarkers to assess disease development. This would therefore improve its management and stop its progression to reduce its morbidity and mortality’’ ''... DN and RNFL loss in turn cause functional modifications in several diagnostic tests that can be detected prior to the appearance of the first manifestations of DR, including electroretinogram [8], contrast sensitivity, dark adaptation, and microperimetry alterations [9–12]. The latter is especially useful for evaluating macular functionality due to its high sensitivity and specificity…’’ or ‘’…The purpose of this study was to measure macular changes in the IRL thicknesses studied by swept source OCT (SS-OCT) in type 2 diabetes mellitus (DM2) patients with moderate DR and without DME compared to that of healthy control individuals and to correlate it with retinal function evaluated by microperimetry to identify markers of retinal neurodegeneration prior to the appearance of DR'') and new references has been included to support the text:
- ''Williams R, Airey M, Baxter H, Forrester J, Kennedy-Martin T, Girach A. Epidemiology of diabetic retinopathy and macular oedema: a systematic review. Eye (Lond). 2004;18(10):963–83.
- ‘’Chhablani J, Sharma A, Goud A, Peguda HK, Rao HL, Begum VU, et al. Neurodegeneration in Type 2 Diabetes: Evidence From Spectral-Domain Optical Coherence Tomography. Invest Ophthalmol Vis Sci. 2015;56(11):6333–8’’.
- ‘’Spaide RF, Fujimoto JG, Waheed NK, Sadda SR, Staurenghi G. Optical coherence tomography angiography. Prog Retin Eye Res. 2018; 64:1–55’’
- ‘’Fernández-Espinosa G, Orduna-Hospital E, Boned-Murillo A, Diaz-Barreda MD, Sanchez-Cano A, Sopeña-Pinilla M, et al. Choroidal and Retinal Thicknesses in Type 2 Diabetes Mellitus with Moderate Diabetic Retinopathy Measured by Swept Source OCT. Biomedicines. 2022;10(9): 2314.
- ‘’Fernández-Espinosa G, Boned-Murillo A, Orduna-Hospital E, Díaz-Barreda MD, Sánchez-Cano A, Bielsa-Alonso S, et al. Retinal Vascularization Abnormalities Studied by Optical Coherence Tomography Angiography (OCTA) in Type 2 Diabetic Patients with Moderate Diabetic Retinopathy.). 2022; 12(2):379.’’.
- ‘’Midena E, Vujosevic S. Microperimetry in diabetic retinopathy. Saudi J Ophthalmol. 2011;25(2):131-5’’.
- ‘’Orduna-Hospital E, Otero-Rodríguez J, Perdices L, Sánchez-Cano A, Boned-Murillo A, Acha J, et al. Microperimetry and Optical Coherence Tomography Changes in Type-1 Diabetes Mellitus without Retinopathy. Diagnostics. 2021;11(1): 136’’.
- In Figure 2 is not visible ground marked in bold
Figure 2 has been enlarged to be clearly read and the units are detailed in the left side of the image, under each row.
- The literature should be prepared in accordance with the requirements of the Biomedicines journal
Literature has been modified in accordance with the requirements of the Biomedicines journal.
Reviewer 3 Report
Comments and Suggestions for Authors
The authors describe the relationship between parameters assessing retina thickness and macular sensitivity in T2DM patients.
The manuscript and study will potentially benefit from addressing the following issues:
• It is unclear what hypothesis has been tested and which mechanisms of action have been identified beyond a straightforward description of phenomena.
• A critical review of similar, already published data and other relevant experimental conditions is needed.
· The methods used to generate data (technical and experimental controls) are described insufficiently. How were artifacts generating differences in assessments of vision loss and diagnosis of dementia controlled for?
· The assessment of confounding ophthalmological and neurological factors such as Alzheimer’s disease, FTD, and other types of neurodegenerative disorders affecting vision, such as stroke, multiple sclerosis, Parkinson Disease, or tauopathies appears inadequate, and should be independently validated, especially when self-reported.
· The information about
o hormone replacement therapies used by female study participants
o the presence or absence of metabolic syndrome
o when vision assessments for individual patients occurred
o how sub-clinical and clinical retinal neovascularization and DME were identified
o how learning and fatigue effects were assessed for microperimetry data acquisition
appears insufficient.
· Visual assessments appear not to be state-of-the-art without adequate ophthalmological evaluation as the Parinaud scale and their Snellen equivalent as well as self-reported loss of distance visual function are subject to numerous confounding factors common in the targeted age groups, such as scotomas, cataracts, corneal degeneration, and the spectrum of dry eye disease related vision loss.
· The description of statistics is insufficient and additional information regarding group sizes and types of analyses need to be provided.
·
Comments on the Quality of English LanguageThe manuscript should be proofread carefully to eliminate typographical, grammar and syntax errors, as well as the low quality of the figures and tables, that make the manuscript in part difficult to understand.
Author Response
Dear Reviewer,
We appreciate your comments about the interest of this manuscript.
We revised the manuscript according to your suggestion and the last change included in the manuscript is detailed below.
We will try to answer your comments point-by-point.
- Extensive editing of English language required
Manuscript has undergone an extensive English revision as suggested (Please see the attachment).
- It is unclear what hypothesis has been tested and which mechanisms of action have been identified beyond a straightforward description of phenomena.
The paper has been deeply edited to clarify the hypothesis and which mechanisms of action have been identified beyond: ‘’…The purpose of this study was to measure macular changes in the IRL thicknesses studied by swept source OCT (SS-OCT) in type 2 diabetes mellitus (DM2) patients with moderate DR and without (DME compared to that of healthy control individuals and to correlate it with retinal function evaluated by microperimetry to identify markers of retinal neurodegeneration prior to the appearance of DR’’ or ‘’In conclusion, our results suggest that the structure (total retinal, GC-IPL and RNFL thicknesses) and function (retinal sensitivity) of the retina in patients with moderate diabetes display some changes. We demonstrated a correlation between the RNFL and retinal sensitivity measured by MAIA, and findings were not present when evaluating the GCL+. The diagnosis of GCL and RNFL thinning in DM2 patients without DR prior to the appearance of other DR signs would enable an accurate approach to this growing disease, with personalized assessment based on the DR course or stage. Future studies require a larger sample size, which should include patients in different stages of the disease and both DM1 and DM2 patients. Systemic risk factors are expected to be involved in retinal neurodegeneration in DM2 patients. However, the role of these factors remains largely unknown, and future lines of work should be promoted to determine it’’.
- A critical review of similar, already published data and other relevant experimental conditions is needed.
Discussion section of the paper has been rewritten to improve this.
- The methods used to generate data (technical and experimental controls) are described insufficiently. How were artifacts generating differences in assessments of vision loss and diagnosis of dementia controlled for?
We answer this point together with the following point 5.
- The assessment of confounding ophthalmological and neurological factors such as Alzheimer’s disease, FTD, and other types of neurodegenerative disorders affecting vision, such as stroke, multiple sclerosis, Parkinson Disease, or tauopathies appears inadequate, and should be independently validated, especially when self-reported.
In our ophthalmology service there is a very large population of type 2 diabetic patients who undergoes rigorous control of DR signs, since our inclusion criteria were very strict, we were not able to include all of them. Within these inclusion criteria we took into account that they did not suffer from any neurodegenerative disease or signs of it since they also suffer retinal changes (as our research group has studied extensively in other research: multiple sclerosis, Parkinson, Alzheimer, bipolar disorder... ), as well as the influences that their treatments have on their vision, either functional or structural findings.
The patients were referred from the Endocrinology Department, their medical history was deeply reviewed and they were discarded based on their systemic pathologies, diagnosis or signs of neurodegenerative disorders, medications that affect vision or previous ocular surgeries, leaving us with a population of 54 DM2 that met our strict inclusion criteria.
We have included in the inclusion/exclusion criteria: “The electronic medical history of the participants was deeply reviewed. Patients with uncontrolled arterial hypertension, diagnosis of metabolic syndrome, other systemic pathologies or diagnosis or signs of neurodegenerative disorders were also excluded, as patient with hormone replacement therapies.”
- Insufficient information about hormone replacement therapies used by female study participants
Female patients with hormone replacement therapies were excluded from the study and it was indicated in the text ‘’… were also excluded, as patients with hormone replacement therapies’’.
- Insufficient information about the presence or absence of metabolic síndrome
Patients with diagnosis of metabolic syndrome were excluded from the study and it was indicated in the text: ‘’Patients with uncontrolled arterial hypertension or diagnosis of metabolic syndrome were also excluded
- Insufficient information about when vision assessments for individual patients occurred
As suggested, it was clarified that all the examinations were performed in the same visit ‘’at the Ophthalmology Department in the Lozano Blesa University Hospital in Zaragoza, Spain from October 2021 through June 2022’’ ‘’ All subjects underwent a complete ophthalmological exam in a single visit’’.
- Insufficient information about how sub-clinical and clinical retinal neovascularization and DME were identified
We emphasize in the text that any sub-clinical and clinical retinal neovascularization and DME were excluded with wield field retinography and OCT: ‘’wield-field retinography and OCT were used to exclude sub-clinical and clinical retinal neovascularization and DME’’.
- Insufficient information about how learning and fatigue effects were assessed for microperimetry data acquisition
A paragraph about has been added: ‘However, we have to consider the limitations of microperimetry, which include sources of variability such as fatigue and learning. Differences between right and left eye parameters related to fatigue have been described. This is because the duration of the test exceeds 5 minutes, and in convention, the right eye is analysed first [36]. This could be reduced by testing only the study eye or if we perform it before the rest of the tests. Regarding microperimetry learning tests, it is not clear if performing a theme is recommended, as the duration of learning effects is unknown [36] ‘’.
- Visual assessments appear not to be state-of-the-art without adequate ophthalmological evaluation as the Parinaud scale and their Snellen equivalent as well as self-reported loss of distance visual function are subject to numerous confounding factors common in the targeted age groups, such as scotomas, cataracts, corneal degeneration, and the spectrum of dry eye disease related vision loss.
These factors were also considered in the inclusion criteria, we have expanded the section so that they are well specified:
“Exclusion criteria for both groups were amblyopia or best corrected visual acuity (BCVA) lower than 20/40 in the Snellen scale, spherical equivalent (SE) above +/- 5.50 diopters (D) or 3.00 D of astigmatism, intraocular pressure (IOP) over 20 mmHg or findings that suggested glaucoma, other macular diseases with macular impairment that lead to scotomas that affect vision, history of ocular surgery, corneal degeneration or lens opacity, the spectrum of vision loss related to dry eye disease or impossibility to collect good quality OCT profile.”
- The description of statistics is insufficient and additional information regarding group sizes and types of analyses need to be provided.
The description of the statistical analysis carried out has been increased and the sample size calculation has been specified:
“…The parameters had not a normal distribution, so differences between groups were analyzed with the Mann–Whitney U test for independent samples and the Spearman’s rank correlation coefficient test was conducted for bivariate analysis to correlate the variables of interest. A value of p<0.05 indicated statistical significance for all the analyses.
The sample size was calculated based on a preliminary study carried out by our group and using the Epidat software (version 4.2.0.0). All calculations were performed applying a two-sided test with α risk of 5% (95% confidence level) and β risk of 20% (80% power). A standard deviation of retinal thickness of 30 μm was estimated, wanting to detect differences of at least 16 μm and setting a ratio between independent sample sizes of 1.5, a minimum sample size of 71 and 47 subjects per group was determined.”

Round 2
Reviewer 1 Report
Comments and Suggestions for Authors
Dear authors,
thank you very much for your thorough responses to my comments. The manuscript reads now much better.
Comments on the Quality of English LanguageEnglish is fine, minor editing may be required to aid flow
Author Response
Dear Reviewer,
We appreciate your comments about the interest of this manuscript.
We revised the manuscript according to your suggestion and the last change included in the manuscript is detailed below.
We will try to answer your comments point-by-point.
-English is fine, minor editing may be required to aid flow
The paper was reviewed by a language editing company. We looked for the grammar mistakes that you said, and we tried to correct all of them. If you find any other mistakes, please let us now. We really appreciate your input and help with the manuscript.
Reviewer 3 Report
Comments and Suggestions for Authors
The authors responded effectively to some of the reviewers' concerns, but have only partially included their responses in the manuscript. Readers will not be able to see these comments and including them in the methods and results sections is needed.
Comments on the Quality of English LanguageSignificant concerns regarding grammar, spelling and syntax remain (e.g., "wield-field retinography", "as patient with hormone replacement therapies", ...).
Author Response
Dear Reviewer,
We appreciate your comments about the interest of this manuscript.
We revised the manuscript according to your suggestion and the last change included in the manuscript is detailed below.
We will try to answer your comments point-by-point.
The authors responded effectively to some of the reviewers' concerns, but have only partially included their responses in the manuscript. Readers will not be able to see these comments and including them in the methods and results sections is needed.
We underlined the changes that we made in the manuscript and included the totality of them. We apology if we missed something; we hope everything is ok now.
-Significant concerns regarding grammar, spelling and syntax remain (e.g., "wield-field retinography", "as patient with hormone replacement therapies", ...).
The paper was reviewed by a language editing company. We looked for the grammar mistakes that you said, and we tried to correct all of them. If you find any other mistakes, please let us now. We really appreciate your input and help with the manuscript.
Round 3
Reviewer 3 Report
Comments and Suggestions for Authors
The authors responded effectively to the reviewers' concerns.
Comments on the Quality of English LanguageMinor editing of English language would further improve the manuscript